# Potential Ototoxicity of Insulin-like Growth Factor 1 Receptor Signaling Inhibitors: An In Silico Drug Repurposing Study of the Regenerating Cochlear Neuron Transcriptome

**DOI:** 10.3390/jcm12103485

**Published:** 2023-05-16

**Authors:** Lino E. Bertagnoli, Richard Seist, Shelley Batts, Konstantina M. Stankovic

**Affiliations:** 1Department of Otolaryngology—Head and Neck Surgery, Stanford University School of Medicine, Stanford, CA 94305, USA; 2Paracelsus Medical University, 5020 Salzburg, Austria; 3Department of Neurosurgery, Stanford University School of Medicine, Stanford, CA 94305, USA; 4Wu Tsai Neuroscience Institute, Stanford University, Stanford, CA 94305, USA

**Keywords:** adverse events, insulin-like growth factor 1 receptor inhibitor, in silico drug repurposing, ototoxicity, spiral ganglion neurons, teprotumumab

## Abstract

Spiral ganglion neurons (SGNs) connect cochlear hair cells with higher auditory pathways and their degeneration due to drug toxicity (ototoxicity) contributes to hearing loss. This study aimed to identify drug classes that are negatively correlated with the transcriptome of regenerating SGNs. Human orthologs of differentially expressed genes within the regenerating neonatal mouse SGN transcriptome were entered into CMap and the LINCS unified environment and perturbation-driven gene expression was analyzed. The CMap connectivity scores ranged from 100 (positive correlation) to −100 (negative correlation). Insulin-like growth factor 1/receptor (IGF-1/R) inhibitors were highly negatively correlated with the regenerating SGN transcriptome (connectivity score: −98.87). A systematic literature review of clinical trials and observational studies reporting otologic adverse events (AEs) with IGF-1/R inhibitors identified 108 reports (6141 treated patients). Overall, 16.9% of the treated patients experienced any otologic AE; the rate was highest for teprotumumab (42.9%). In a meta-analysis of two randomized placebo-controlled trials of teprotumumab, there was a significantly higher risk of hearing-related (pooled Peto OR [95% CI]: 7.95 [1.57, 40.17]) and of any otologic AEs (3.56 [1.35, 9.43]) with teprotumumab vs. a placebo, whether or not dizziness/vertigo AEs were included. These results call for close audiological monitoring during IGF-1-targeted treatment, with prompt referral to an otolaryngologist should otologic AEs develop.

## 1. Introduction

The sensitivity of human hearing relies on the integrity of the cells of the cochlea, particularly the mechanosensory inner and outer hair cells and the spiral ganglion neurons (SGNs), which transmit sound information to higher auditory processing centers [1]. Mammalian hair cells are tonotopically organized within the spiraling cochlea; a single inner hair cell (IHC) typically synapses with 10 boutons from 10 different spiral ganglion cells, with higher synaptic density (up to 20 per IHC) in the most sensitive frequency regions [2,3]. Cochlear cells may be damaged or lost due to noise, aging, trauma, or drug toxicity (i.e., ototoxicity), leading to hearing impairment that is often permanent [4,5,6]. This is because, unlike other species, adult mammals have no capacity to spontaneously regenerate cochlear hair cells once lost [7,8], and SGN regeneration was historically believed to be limited to neonate mammals [9]. However, some adult mammalian species have demonstrated regenerative capacity in the cochlear cells, which appears to be limited to fibrocytes of the spiral ligament, afferent synapses, and the peripheral axons of SGNs [10,11,12,13]. Additionally, many of the molecular pathways and growth factors directing SGN regeneration in neonate mammals are also implicated in the maintenance and survival of mature SGNs and their synapses with hair cells [9,14,15,16,17]. For this reason, understanding the mechanism of SGN regeneration in neonates—as well as potential disruptors of this process—may yield clues as to how to preserve SGNs in adults exposed to noise or chemicals that are toxic to cochlear cells, or who are subject to age-related hearing decline.

Certain drugs are known to be ototoxic and to cause excitotoxic damage, functional impairment, and the degeneration of inner ear cells such as SGNs, including platinum-based chemotherapeutics, aminoglycoside antibiotics, loop diuretics, and non-steroidal anti-inflammatory drugs [6,18,19]. As the process of drug discovery advances, novel compounds and classes have been revealed to have ototoxic effects, such as some newer non-platinum-based chemo- and immunotherapies [18,20]. An example of a novel, potentially ototoxic drug class includes inhibitors of insulin-like growth factor 1 and its receptor (IGF-1/R) [21,22]. IGF-1 is an important endocrine hormone that promotes bone and tissue growth as a mediator of growth hormone [22]. It is also one of the most potent activators of the anaplastic lymphoma kinase (ALK) signaling pathway, which stimulates cell growth and inhibits apoptosis [23]. However, alterations in *ALK* can lead to an oncogenic form of the gene and cancer development [24]. For these reasons, inhibitors of IGF-1/R have been assessed as therapies for various cancers in clinical trials, although, as of 2023, only two—ceritinib and teprotumumab—have been approved by the United States Food and Drug Administration (FDA) for any indication [25,26]. Ceritinib (an off-target IGF-1R antagonist) was approved by the FDA in 2014 for the treatment of ALK-mutated non-small cell lung cancer, while teprotumumab (an IGF-1R antagonist) was approved in 2020 for the treatment of active thyroid eye disease (TED, also known as Graves’ ophthalmopathy) [25,26]. The approval of teprotumumab was based on its efficacy, as demonstrated in two clinical trials (NCT01868997 and NCT03298867) wherein 170 total patients with TED were randomized to teprotumumab infusions (*n* = 84) or placebo (*n* = 87) [27,28]. In both trials, significantly more teprotumumab-treated patients achieved a ≥ 2 mm reduction in proptosis (an ocular symptom of TED) compared with placebo (83–71% vs. 10–20%). However, there were unexpected otologic adverse events (AEs), such as hearing loss and tinnitus, associated with teprotumumab in the clinical trials, which have continued to be reported in real-world clinical practice [21,29,30].

The mechanism underlying the potential ototoxicity of IGF-1/R inhibitors is currently unknown but may be related to the disruption of an ongoing neuroprotective role of IGF-1 in the cochlea. The *Igf1* gene is expressed in the mammalian cochlea during inner ear development in a region-specific pattern but persists at lower levels into adulthood [31,32]; in mice, *Igf*-1 is located in SGNs and the stria vascularis [33]. Without IGF-1, cochlear development is delayed or abnormal, and inborn IGF-1 deficiency results in profound sensorineural hearing loss in humans and mice [31,32,34,35,36]. Thus, the perturbation of IGF-1 signaling, as with the use of IGF-1/R inhibitors, may negatively impact the function or survival of mature SGNs, leading to the onset of otologic AEs such as hearing loss or tinnitus.

It is imperative to weigh the therapeutic benefits of a drug with the potential risks of irreversible hearing loss due to ototoxicity, although ototoxic AEs may not always appear in the context of a clinical trial conducted among strictly selected patient populations. It is also important to reduce the risk of harm from AEs in patients, both when considering the clinical applications of new drugs and in the context of post-approval monitoring. Accordingly, in silico methods to screen for potentially ototoxic drug classes are valuable tools to inform treatment decisions and to avoid the emergence of AEs during clinical trials and in real-world clinical practice [4,20]. Such methods include digital drug-repurposing hubs, which contain information on approved, pre-clinical, and experimental compounds, and allow screening to determine new therapeutic targets for or illuminate potential adverse reactions to existing drugs [37,38,39].

Therefore, to improve our understanding of the potential ototoxicity of IGF-1/R inhibitors, as well as other classes of drugs, we employed the drug repurposing suite CLUE (Connectivity Map (CMap) and the Library of Integrated Network-based Cellular Signatures (LINCS) unified environment) to analyze connections in the perturbation-driven gene expression dataset for an in silico screen of regenerating mammalian auditory neurons. Specifically, human orthologs of murine genes with differential expression during SGN damage and subsequent regeneration, as reported by Wu et al. [14] using neonatal mouse cochlear explants exposed to kainic acid, were correlated with the putative ototoxic effects of drug classes. Neonatal cochlear explants provide a good model for assessing the drug sensitivity of the adult cochlea [40,41], and kainic acid exposure is a well-established method for inducing excitotoxic damage to auditory neurites that is similar to damage resulting from drugs or loud noise [42,43]. Furthermore, we conducted a systematic literature review of the reports of ototoxic AEs associated with IGF-1/R inhibitors, along with a preliminary meta-analysis of randomized, placebo-controlled trials of teprotumumab.

## 2. Materials and Methods

### 2.1. Data Source

Publicly available microarray data of afferent SGN neuron regrowth after excitotoxic injury were analyzed (data accessible at the National Center for Biotechnology Information (NCBI) Gene Expression Omnibus database (Wu et al., 2020), accession GSE130495) [14]. In the research reported by Wu et al. [14], neonatal murine cochlear explants were exposed to 0.5 mM kainic acid for 2 h to induce excitotoxicity and SGN retraction, followed by subsequent regeneration after 24 h (Figure 1). Therefore, the microarray data of SGN tissue corresponding to 5 h (retraction phase) and 24 h (regeneration phase) after excitotoxic injury with kainic acid were selected for analysis, representing the time of afferent neuron regrowth. Retraction and regeneration were histologically confirmed by Wu et al. via staining with antibodies against anti-neurofilament H (SGNs), C-terminal binding protein 2 (pre-synaptic ribbons), and post-synaptic density protein 95 (postsynaptic patches) [14]. As this study used previously published, publicly available data, no institutional board review was required.

### 2.2. In Silico Screen

This study employed GEO2R, an interactive web tool that identifies differences in gene expression across experimental conditions (https://www.ncbi.nlm.nih.gov/geo/geo2r/; NCBI) (accessed on 1 November 2021), and the web-based drug repurposing suite, CLUE (CMap and LINCS unified environment; https://clue.io/; Broad Institute, MIT) (accessed on 1 November 2021) [38,44]. CMap has generated signatures from genetic loss-of-function and overexpression mutants, covering over 7000 genes associated with human disease or the involved biological pathways and processes, and holds the largest set of data on drug-induced gene expression changes in human cell lines [38,39]. The signatures are based on Affymetrix microarrays from ~1300 small molecule treatments (L1000 assay), where changes in gene expression are measured following treatment with a perturbagen (i.e., groups of compounds that share the same mechanism of action or biological function).

The microarray data from GSE130495 were first analyzed using GEO2R to identify murine genes that were differentially expressed during the retraction and regeneration phases reported by Wu et al. Then, the human orthologs for the differentially expressed genes (DEGs) were inputted to CLUE to analyze connections in the perturbation-driven gene expression dataset of regenerating SGNs. Of the 63 DEGs identified, 4 of the 26 up-regulated DEGs and 6 of the 37 down-regulated DEGs were excluded from the CLUE query as they either had no Human Genome Organization symbol, had no valid Entrez identification, or were not part of the Best INFerred Genes feature space (Appendix A). The selected DEGs during the SGN regeneration phase were analyzed by CLUE for perturbagen classes with similar or opposite signatures. CMap signatures were given a connectivity score of between 100 (positive correlation) and −100 (negative correlation), with 0 representing no correlation. Scores over 90 or under −90 were considered highly correlated [39].

### 2.3. Systematic Review of Publications Reporting Otologic AEs Associated with IGF-1/R Inhibitors

A systematic literature review was performed in PubMed in August 2022 and in ClinicalTrials.gov in January 2023 to identify English-language publications and clinical trials published between January 2005 and August 2022 describing otologic AEs of drugs targeting IGF-1/R. Search terms in PubMed included “teprotumumab”, “R1507”, “Tepezza”, “cixutumumab”, “IMC-A12”, “ganitumab”, “AMG-479”, “figitumumab”, “CP-751,871”, “dalotuzumab”, “MK-0646”, “ceritinib”, “LDK378”, “Zykadia”, “ganetespib”, “STA-9090”, or “linsitinib”, combined with “hearing”, “hearing loss”, “deaf”, “deafness”, “tinnitus”, “vertigo”, “ear pain”, “ear discomfort”, “hypoacusis”, “ototoxic”, “ototoxicity”, “dizziness”, “labyrinth”, “audiometry”, or “audiogram”. Furthermore, publications identified from a PubMed title/abstract search (in August 2022) with search terms including “cixutumumab”, “ganitumab”, “figitumumab”, “dalotuzumab”, “ganetespib”, “linsitinib”, or “teprotumumab” were manually reviewed for otologic AEs (e.g., hearing loss or tinnitus). Clinical trials registered in ClinicalTrials.gov were also screened for inclusion of the drugs listed above. Trials with reported results were scanned for reports of otologic AEs.

The PRISMA flowchart illustrated in Figure 2 details the search process and Appendix A Appendix A lists the search terms and pharmacological targets of the drugs. Publications and trials with overlapping results were analyzed together and summarized to extract the best possible data and avoid duplicate records. Although R1507 is also sometimes listed as the experimental name of teprotumumab, it was included separately in the literature search as it was not possible to determine whether alterations had occurred in the molecule prior to the market entry of teprotumumab.

The results from the literature review were then compiled, including the drug name, sample size (*n*), and the numbers of total otological AEs and otologic AEs by type. Because the reports had various methods of reporting AEs, otologic AEs were categorized into hearing AEs, tinnitus AEs, other otologic AEs, and dizziness and/or vertigo events, as follows. Terms categorized as “hearing AEs” included hypoacusis, deafness, unilateral deafness, bilateral deafness, neurosensory deafness, hearing impaired, hyperacusis, muffled hearing, hearing loss, diminished word recognition, and abnormal audiogram. Terms categorized as “tinnitus AEs” included tinnitus and ear popping, while terms categorized as “dizziness and vertigo” included dizziness, vertigo, balance disorder, motion sickness, and positional vertigo. “Other otologic AEs” included ear discomfort, ear plugging, fullness and pressure, autophony, patulous Eustachian tube, ear or labyrinth disorders, ear congestion, cerumen impaction, ear infection, ototoxicity, tympanic membrane perforation, ear disorder, middle ear effusion, external ear inflammation, labyrinthitis, and auditory/ear–other AEs.

### 2.4. Meta-Analysis of Randomized Placebo-Controlled Trials of Teprotumumab for TED

A meta-analysis of randomized, placebo-controlled clinical trials of teprotumumab for the treatment of TED was performed to compare the rates of otologic AEs associated with teprotumumab versus a placebo. Data were extracted from two eligible randomized controlled trials of teprotumumab (NCT01868997 (Smith et al. [27]) and NCT03298867 (Douglas et al. [28])), involving a total of 170 patients with TED from both trials. NCT01868997 was a Phase 2 clinical trial, with 43 participants in the treatment group and 44 in the placebo group [27]. NCT03298867 was a Phase 3 clinical trial, with 41 participants in the treatment group and 42 in the placebo group [28]. In both trials, teprotumumab-treated patients received the same cumulative dose: one infusion was given every 3 weeks, with the first at 10 mg/kg and the remaining 7 at 20 mg/kg [45,46]. Otologic AEs recorded in these trials were evaluated and categorized into hearing, tinnitus, dizziness/vertigo, and other otologic AEs as described above. A risk-of-bias analysis was performed using the Review Manager software (RevMan v5.4, Cochrane, London, UK) for the trials selected for inclusion in the meta-analysis (Appendix A Appendix A).

### 2.5. Statistical Analyses

The rates of otologic AEs calculated in the systematic literature review were reported for each drug overall and according to AE type as counts and proportions. As an FDA-approved IGF-1/R therapy in current clinical use, teprotumumab was of special interest. Therefore, the rates of otologic AEs reported in the clinical trials of teprotumumab were also compared with the real-world rates from an observational study by Sears et al. (2022) [21].

In the meta-analysis, the rates of otologic AEs, both overall and by type, were compared between the teprotumumab and placebo groups using the Review Manager software and were summarized as Peto odds ratios (OR) along with their corresponding 95% confidence intervals (CI). The Peto method was chosen because it has been observed to be less biased and more powerful than other methods when looking at rare events in small groups [47]. Because AEs were not always described in detail, preventing the ability to discern if dizziness/vertigo events were always of otologic origin, comparisons in the meta-analysis were conducted, both including and excluding dizziness/vertigo events. Statistical heterogeneity was assessed using the I^2^ and chi-square tests. Both outcome measures were calculated for the duration of the underlying trials.

## 3. Results

### 3.1. CLUE Query of Drug Classes Negatively Correlated with SGN Regeneration

In the CLUE query, six CMap drug compound classes were found to be highly negatively correlated with SGN regrowth and synaptogenesis (i.e., having a connectivity score lower than −90). These included inhibitors of IMPDH (connectivity score: −98.91), IGF-1 (−98.87), mTOR (−97.46), PI3K (−96.33), DNA-dependent protein kinase (−92.91), and HMGCR (−90.86) (Table 1).

### 3.2. Systematic Literature Review

Given the strong negative correlation between IGF-1/R inhibitors and SGN regeneration in the CLUE query, a literature review was conducted to assess the reported rates of otologic AEs associated with experimental or FDA-approved IGF1/R inhibitors. Of the 36 eligible publications and 72 eligible clinical trials identified, 36 publications and trials had overlapping results and were consolidated to avoid duplication. In total, otologic AE data from 108 reports, representing 6141 patients treated with IGF1/R inhibitors, were extracted. An application document for FDA approval of teprotumumab (application number 761143Orig1s000 [48]) was also incorporated. Of the 6141 total patients treated with IGF-1/R inhibitors, 2825 received monotherapy, 3315 received combination therapy, 5951 were treated for cancer, 161 were treated for TED, and 28 were healthy.

Across all the IGF-1/R inhibitors, 16.9% of treated patients experienced any otologic AEs, including 1.6% with hearing AEs, 1.7% with tinnitus AEs, 11.4% with dizziness or vertigo AEs, and 2.2% with other otologic AEs (Table 2). The overall rate of otologic AEs was highest among patients treated with teprotumumab, where 42.9% (*n* = 69/161) experienced any otologic AEs. Similarly, the rates of hearing AEs (13.0%; *n* = 21/161), tinnitus AEs (8.7%; *n* = 14/161), and other otologic AEs (16.8%; *n* = 27/161) were all highest among teprotumumab-treated patients. The highest incidence of dizziness and vertigo was among ganitumab-treated patients (14.9% (*n* = 38/255)). For a comparison of the publications and trials of teprotumumab, refer to Appendix A.

### 3.3. Meta-Analysis of Otologic AEs Reported in Placebo-Controlled Trials of Teprotumumab

#### 3.3.1. Total Otologic AEs

In the meta-analysis of placebo-controlled teprotumumab trials, teprotumumab was associated with a higher risk of any otologic AE compared with placebo when either including or excluding dizziness/vertigo as an event (pooled Peto OR [95% CI]: 3.56 [1.35, 9.43] and 8.27 [2.35, 29.83], respectively) (Figure 3 and Table 3). In both cases, the overall effect was significant, with a Z-score of 2.56 (*p* = 0.01) when including dizziness/vertigo and 3.28 (*p* = 0.001) when excluding it.

The trials displayed moderate heterogeneity when including dizziness/vertigo (chi^2^ statistic: 2.95; degrees of freedom (df) = 1, *p* = 0.09; I^2^ = 66%) and low heterogeneity when excluding it (chi^2^ statistic: 0.00; df = 1, *p* = 1.00; I^2^ = 0%).

#### 3.3.2. Otologic AEs by Type

Compared with a placebo, teprotumumab was associated with a higher risk of hearing-related AEs (pooled Peto OR [95% CI]: 7.95 [1.57, 40.17]) and the effect was significant, with a Z-score of 2.51 (*p* = 0.01) (Figure 3 and Table 3). Teprotumumab was not associated with a higher risk of dizziness/vertigo (pooled Peto OR [95% CI]: 1.02 [0.25, 4.21]; Z-score: 0.03, *p* = 0.97). No tinnitus events were reported in NCT03298867 and no “other” otologic AEs were reported in NCT01868997. Thus, teprotumumab may be associated with a higher risk of tinnitus (pooled Peto OR [95% CI]: 7.94 [0.49, 129.15]) and other otologic AEs (7.76 [0.48, 126.27]) vs. placebo, although the overall effect was not significant in either case (Z = 1.46 and 1.44, respectively; both *p* > 0.05). Additionally, the wide CIs indicate a large degree of uncertainty around the OR estimates.

The trials displayed low heterogeneity when assessing hearing AEs (chi^2^ statistic: 0.00; df = 1, *p* = 1.00; I^2^ = 0%) and high heterogeneity when assessing dizziness (chi^2^ statistic: 4.97; df = 1, *p* = 0.03; I^2^ = 80%). Heterogeneity was not estimable in the assessment of tinnitus and other otologic AEs.

## 4. Discussion

This in silico screen of regenerating auditory neurons demonstrated that IGF-1/R inhibition was highly inversely correlated with the transcriptome of mammalian SGNs during regrowth and synaptogenesis, suggestive of the ototoxic capacity of the drug class. This discovery informed a systematic literature review to summarize the rates of otologic AEs that have been reported for IGF-1/R inhibitors in clinical trials and real-world observational studies, determined to be 17% across the drugs but 43% for teprotumumab. A small meta-analysis of the two randomized, placebo-controlled trials of teprotumumab indicated that teprotumumab was associated with a significantly higher risk of hearing-related and overall otologic AEs compared with a placebo (including or excluding dizziness as an event).

IGF-1 is known to be generally neuroprotective and its activity aids in maintaining cellular metabolism, promoting growth, influencing cell proliferation and differentiation, and inhibiting apoptosis [22,49,50]. Furthermore, IGF-1 promotes peripheral nerve elongation and branching and the activation of stem cells during regenerative processes (i.e., in muscles) [51,52]. The results from the in silico screen underscore the important role of IGF-1 in the inner ear, which is supported by prior observations in the literature. IGF-1R is expressed on the surface of mammalian inner and outer hair cells, all supporting cells, and in SGNs, and is crucial for directing the proper differentiation of otic precursors during development [53,54]. In humans, IGF-1 deficiency is a rare condition associated with sensorineural hearing loss, poor growth rates, and cognitive impairment [34,35,36]. Similarly, *Igf1*−/− mice are dwarfs with poor survival rates and bilateral profound hearing loss [55], while mice with reduced production of IGF-1 (*Igf1*−/+) do not show initial hearing loss but are more susceptible to noise-related hearing loss (NIHL) than normal mice [32]. Interestingly, there is a case report of exacerbated hearing loss after blast trauma (unprotected gunshot) in a patient under teprotumumab therapy [56], although future research is needed to examine the causality. In animal models, IGF-1 has shown a capacity for use as an otoprotective molecule against NIHL [57,58], aminoglycoside toxicity [59], ischemic injury [60], and surgical trauma from cochlear implantation [61].

Although there is limited direct evidence demonstrating that monoclonal antibodies that are delivered systemically reach the cochlea, antibodies against supporting cells have been shown to cause ototoxicity after systemic injection, suggesting that antibodies may be able to cross the blood-labyrinthine barrier [62,63]. Additionally, synaptic damage caused by the intracochlear perfusion of tumor necrosis factor-alpha (TNF-α) can be prevented by systemic administration of the TNF-α blocker etanercept, a fusion protein similar in size to monoclonal antibodies [64]. Moreover, the blood labyrinthine barrier has dynamic permeability, which can be affected by inflammation and certain disease processes [65,66]. Future research should focus on the pharmacokinetics of monoclonal antibodies, as there is a great need to understand how these antibodies are distributed in the body. Considering this evidence along with the results of the in silico screen helps to explain why prolonged systemic inhibition of IGF-1 or its receptor, as during the treatment of TED with teprotumumab, may present with ototoxic AEs [67].

It is worth noting that the rates of otologic AEs reported in the clinical trials of teprotumumab (pre- and post-FDA approval for TED) were substantially lower than those reported in a recent real-world observational study by Sears et al. [21] (Appendix A Appendix A), underscoring the need for the ongoing monitoring of otologic AEs in the clinic. In the Phase 2 (NCT01868997) [27,45] and Phase 3 (OPTIC; NCT03298867) [28,46] trials comparing teprotumumab with a placebo for TED, the combined incidence of otologic AEs was 11.9%. A Phase 3 open-label extension study (OPTIC-X; NCT03461211) [68] evaluated teprotumumab among patients in OPTIC who were either proptosis non-responders at Week 24 or who were responders but relapsed during the follow-up period. The incidence of otologic AEs in that trial was 21.6%, including one patient that experienced neurosensory deafness during follow-up. However, Sears et al. [21] reported that 22 out of 27 teprotumumab-treated patients experienced a total of 41 otologic AEs (incidence: 151.9%). This suggests that the present findings regarding the incidence of otologic AEs may be underestimated, given the current immaturity of real-world clinical evidence for teprotumumab. The meta-analyses point to the potential ototoxicity of teprotumumab in its clinical trials, but the recent real-world evidence and case studies suggest the need for long-term post-trial monitoring and the consideration of audiometric screening prior to and during teprotumumab therapy. While the teprotumumab trials’ endpoints were focused on efficacy for TED, practicing clinicians must also weigh individual patients’ other medical needs and set expectations for the benefits and risks of therapy. Thus, there is an important role for the clinician in patient education as well as when considering routine audiograms to detect and evaluate early hearing loss.

However, it is also important to acknowledge that the study designs, patient populations, outcomes of interest, study periods, and identification of AEs vary greatly between clinical trials and observational studies. Trials are conducted prospectively and are selective in terms of the patient pools; participants are closely watched at all times for the development of an AE, per stringent reporting protocols. The identification and definition of otologic AEs may be stricter in a trial (i.e., relying on expert clinical opinion) as compared with real-world studies that may also use patient self-reporting. Thus, while this study makes some limited comparisons of real-world evidence with trial evidence, it is presented with the caveat that only a robust indirect comparison (i.e., using matching-adjusted methods) could control for differences in patient populations or study design. Regardless, there is a need for further research to better understand the mechanisms leading to these otologic AEs during treatment with IGF-1/R inhibitors, which can inform treatment decision-making, the design of future clinical trials, and strategies to minimize the AEs.

This study is subject to several limitations, some of which are inherent to literature reviews and meta-analyses describing rare events. First, CMap relies on transcriptomics analysis from human cancer cell lines and since the dataset from Wu et al. contained murine microarray data, not all genes could be entered into CMap. Although the algorithm has been utilized for drug repurposing in neurological diseases [69], this platform might not be the most optimal choice. For example, a study that compared the transcriptome after treatment with a select number of compounds in neural progenitor cells and differentiated neurons to the transcriptome in cancer cell lines found that the neuronal lines differed more from cancer cell lines than did different cancer cell lines from one another [38]. Thus, the CLUE query of drug classes that are negatively correlated with SGN regeneration and the calculated connectivity scores may be weakened by the limitations of the CMap drug repurposing suite. Second, it is also assumed that there will be inherent genetic differences in the transcriptomes of humans and mice (the reference), namely, because SGN regeneration has not been documented in adult humans. This capacity is documented in early postnatal mice (whose inner ear development stage more closely correlates with late embryonic development in humans) and in adult guinea pigs [12,13]. Importantly, SGN regeneration pathways in neonates contribute to SGN survival throughout life [9,14,15,16,17]. Future studies are recommended to assess the adverse effects of IGF-1/R inhibitors on other cochlear cells required for hearing, such as sensory hair cells. Third, it was often difficult to qualify and quantify the reported AEs in publications and trials because few details were provided regarding the specific attributes of potentially otologic AEs. For example, without further evaluation, events such as dizziness may or may not be of otologic origin. Fourth, the results of the meta-analyses should be interpreted with caution, given the low number of patients treated, which is typical for studies of rare disorders but nevertheless limits statistical power and can lead to wide CIs. Similarly, the meta-analyses included only two eligible trials, which may lead to bias, particularly if the original studies were underpowered. An analysis of bias of the two trials indicated that there was a high risk of bias related to the blinding of outcome assessment data, selective reporting bias, and other bias in both trials (Appendix A). More studies with greater sample sizes and specific otologic screening are needed to confirm the present findings. Alternatively, future research may focus on the use of indirect treatment comparison methods to robustly compare the rates of otologic AEs reported in real-world studies with those reported for trials.

## 5. Conclusions

Ototoxicity is a serious and often irreversible AE related to numerous drugs, and the cascade of events leading to cochlear or vestibular dysfunction is poorly understood. This in silico screen of potentially ototoxic drug classes identified IGF-1/R inhibitors as perturbagens that are strongly negatively correlated with mammalian SGN regeneration following excitotoxic injury. A literature review summarizing the rates of otologic AEs reported in previous studies of IGF-1/R inhibitors identified teprotumumab as having the highest rates across this class, which were significantly higher than for a placebo—for both hearing AEs and overall AEs— in a small meta-analysis of its pivotal trials. In line with these findings, it is recommended to conduct audiometric screening of patients treated with teprotumumab, including baseline audiograms, repeat testing, and consultation with an otolaryngologist/audiologist, should otologic AEs develop.

## Figures and Tables

**Figure 1 jcm-12-03485-f001:**
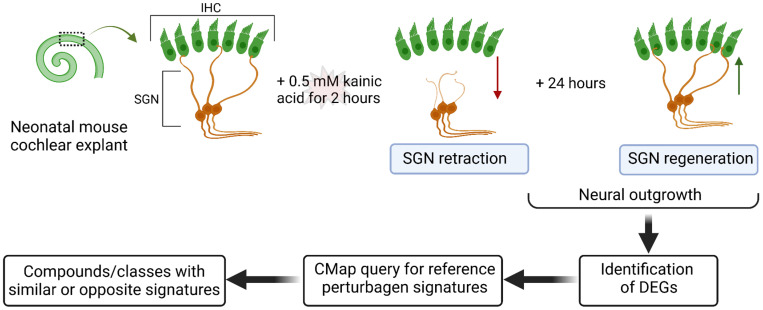
Identification of drug classes that are perturbagens of SGN regeneration. As reported by Wu et al. (top row), neonatal murine cochlear explants were exposed in vitro to 0.5 mM kainic acid for 2 h to cause excitotoxicity and SGN retraction (red arrow) [14]. Over the next 24 h in culture, SGNs grew back toward and synapsed onto IHCs (green arrow). In this study, DEGs during the neural outgrowth phase were analyzed with CMap and the LINCS unified environment (CLUE) to identify perturbagen classes with similar or opposite signatures to the transcriptome of regenerating SGNs. In the human cochlea, each hair cell receives approximately 10 SGN connections and we have presented a simplified representation. Abbreviations: CMap, Connectivity Map; DEG, differentially expressed gene: IHC, inner hair cell; LINCS, Library of Integrated Network-based Cellular Signatures; SGN, spiral ganglion neurite. Created with BioRender (https://biorender.com; accessed on 22 February 2023).

**Figure 2 jcm-12-03485-f002:**
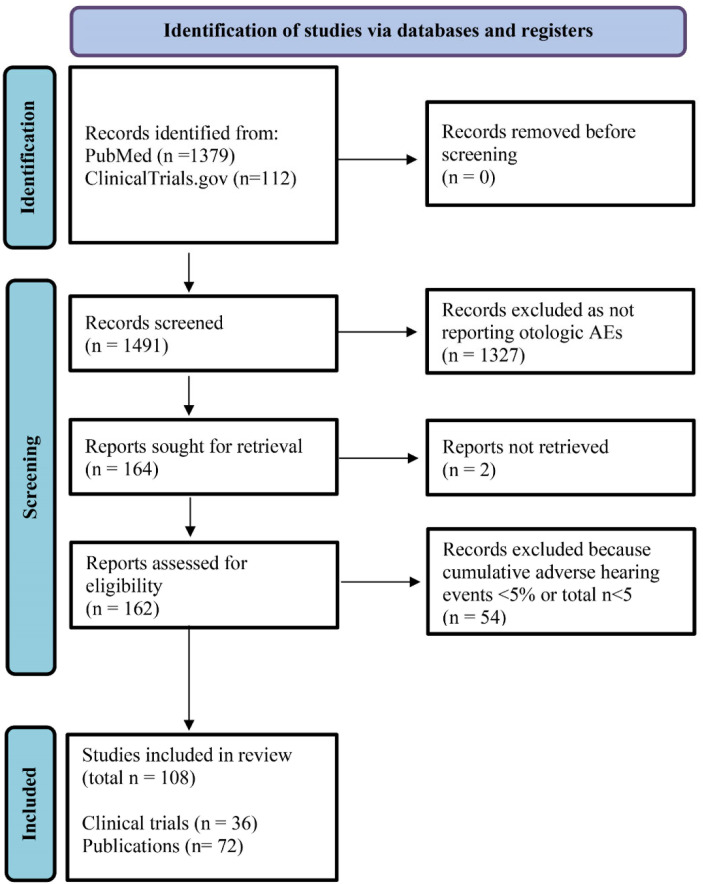
PRISMA flow diagram of the literature review. A literature review of PubMed (for publications) and ClinicalTrials.gov (for registered trials) was conducted to identify the reported rates of otologic adverse events (AEs) associated with drugs targeting insulin-like growth factor 1. Search terms included the drugs’ generic, experimental, and trade names (see Methods Section 2.3). During screening, 36 reports with overlapping results were consolidated.

**Figure 3 jcm-12-03485-f003:**
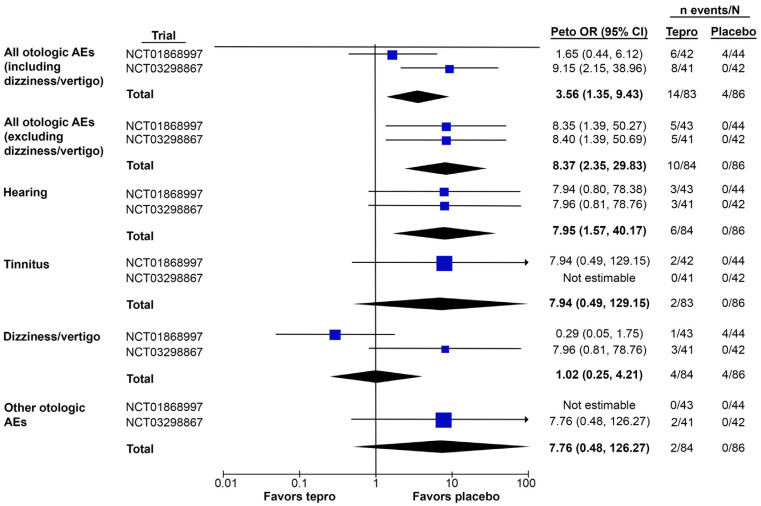
Meta-analysis of the otologic AEs reported in placebo-controlled trials of teprotumumab for TED, both overall and by AE type. ORs and their 95% CIs are represented by blue squares with a horizontal line (individual trials) or black rhombuses (total for both trials). Otologic AEs reported in the placebo-controlled trials of teprotumumab (ClinicalTrials.gov identifiers: NCT01868997 and NCT03298867) were categorized as hearing-related, tinnitus, dizziness/vertigo, or other, as described in Section 2.3. Because dizziness/vertigo could not be definitively attributed to an otologic AE, the meta-analysis of overall AEs was performed, both including and excluding dizziness/vertigo. Abbreviations: AE, adverse event; CI, confidence interval; OR, odds ratio; TED, thyroid eye disease; tepro, teprotumumab.

**Table 1 jcm-12-03485-t001:** The top inversely correlated CMap perturbagen classes during spiral ganglion neurite regeneration.

Drug Class	CMap Connectivity Score
IMPDH inhibitor	−98.91
IGF-1 inhibitor	−98.87
mTOR inhibitor	−97.46
PI3K inhibitor	−96.33
DNA-dependent protein kinase inhibitor	−92.91
HMGCR inhibitor	−90.86

Connectivity scores range from 100 to −100. Scores over 90 or under −90 are considered highly correlated. Abbreviations: CMap, Connectivity Map; HMGCR, 3-hydroxy-3-methyl-glutaryl-coenzyme A reductase; IGF-1, insulin-like growth factor 1; IMPDH, inosine monophosphate dehydrogenase; mTOR, mammalian target of rapamycin; PI3K, phosphoinositide 3-kinase.

**Table 2 jcm-12-03485-t002:** Summary of the otologic AEs associated with drugs targeting IGF-1, as reported in the records identified in the literature review.

	Publication/Trial Data		Reported Otologic AEs, *n* (%)				
Drug	N of Trials/Publications	Total N of Patients	Hearing	Tinnitus	Dizziness/Vertigo	Other	Total
Teprotumumab ^a^	3/4	161	21 (13.0%)	14 (8.7%)	7 (4.3%)	27 (16.8%)	69 (42.9%)
R1507	4/2	473	0 (0%)	5 (1.1%)	25 (5.3%)	11 (2.3%)	41 (8.7%)
Cixutumumab	26/10	1566	14 (0.9%)	24 (1.5%)	210 (13.4%)	32 (2%)	280 (17.9%)
Ganitumab	4/1	255	3 (1.2%)	4 (1.6%)	38 (14.9%)	4 (1.6%)	49 (19.2%)
Figitumumab	9/5	1389	48 (3.5%)	32 (2.3%)	142 (10.2%)	38 (2.7%)	260 (18.7%)
Dalotuzumab	9/6	616	9 (1.5%)	4 (0.6%)	57 (9.3%)	8 (1.3%)	78 (12.7%)
Ceritinib ^a^	10/7	1253	4 (0.3%)	19 (1.5%)	185 (14.8%)	10 (0.8%)	218 (17.4%)
Ganetespib	7/1	428	2 (0.5%)	2 (0.5%)	35 (8.2%)	3 (0.7%)	42 (9.8%)
**Total**	72/36	6141	101 (1.6%)	104 (1.7%)	699 (11.4%)	133 (2.2%)	1037 (16.9%)

No publications reporting otologic AEs for linsitinib were identified in the search. Abbreviations: AE, adverse event; IGF-1, insulin-like growth factor 1. Note: ^a^ Approved by the United States Food and Drug Administration.

**Table 3 jcm-12-03485-t003:** Statistical results of the meta-analyses of otologic AEs in placebo-controlled trials of teprotumumab for TED, both overall and by AE type.

	Heterogeneity				Overall Effect	
Otologic AE	Chi^2^	df	*p*	I^2^	Z-Score	*p*
All otologic AEs						
Including dizziness/vertigo	2.95	1	0.09	66%	2.56	0.01 *
Excluding dizziness/vertigo	0.00	1	1.00	0%	3.28	0.001 *
Hearing	0.00	1	1.00	0%	2.51	0.01 *
Tinnitus	NE				1.46	0.15
Dizziness/vertigo	4.97	1	0.03	80%	0.03	0.97
Other otologic AEs	NE				1.44	0.15

* *p* < 0.05. Teprotumumab was associated with a significantly higher risk of any otologic AE, including or excluding dizziness/vertigo, and of hearing-related AEs, when compared to placebo in its pivotal clinical trials for TED (ClinicalTrials.gov identifiers: NCT01868997 and NCT03298867). Odds ratios are listed in Figure 3. Abbreviations: AE, adverse event; df, degrees of freedom; NE, not estimable; TED, thyroid eye disease.

## Data Availability

The authors attest that all study data are included in the article or online Appendix A.

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
