# Peer review of "Potential Ototoxicity of Insulin-like Growth Factor 1 Receptor Signaling Inhibitors: An In Silico Drug Repurposing Study of the Regenerating Cochlear Neuron Transcriptome"

_jcm, 2023, doi:10.3390/jcm12103485_

Round 1

Reviewer 1 Report

The manuscript by Bertagnoli et al. uses a combination of in-silico transcriptomic analyses and a systematic review with meta-analysis to identify ototoxic adverse effects (AE) of immunotherapies using insulin-like growth factor 1/receptor inhibitors (IGF-1/R). The manuscript addresses an important problem using a novel approach, the methodology is sound, and the results are appropriately interpreted. There are a few concerns for the authors to address/consider.

First, the authors attention to IGF-1/R is based on an in-silico screening of drug classes that disrupt processes involved in SGN regeneration (regrowth and synaptogenesis) identified from murine microarray data. The rationale motivating this strategy to identify drug classes needs to be provided explicitly. Specifically, what is the rationale to investigate pathways involved in SGN regeneration following a very specific cochlear trauma (administration of kainic acid) to reveal drugs with potential otologic AE? Do the authors suspect there are shared mechanisms interlinking drug-induced ototoxicity with reduced SGN repair and regeneration (even though repair mechanisms are limited to neonates as the authors point out, page 1)? This rationale would motivate the utility of the in-silico analysis and improve cohesion between the in-silico analyses and the subsequent systematic review and meta-analysis. Similarly, although the authors acknowledge the limitation arising from genetic differences between mouse and human (page 11), the authors should also indicate other potential limitations in their screening approach, which did not per se investigate processes related to sensory hair cells (which the authors indicate are also key players in the pathology of ototoxicity, page 1)?

Second, the data presented in Figure 3 and 4 would be more appropriately displayed in a single Table (that includes statistical data) with a separate single figure (that includes graphical data with odds ratios).

Third, the authors should consider assessing publication bias and including a description of the potential consequences of publication bias for the interpretation of their results.

Fourth, the authors should address possible caveats of a meta-analysis including only two studies, especially if the individual studies might be under-powered.

Reviewer 2 Report

This is an interesting study using data in transcriptome of mouse cultures and clinical trials associated with IGF1/R inhibitors, which is worthy for publication in jcm. There are several concerns before publication.

Major critiques

1) The title can mislead the readers. A term of insulin-like growth factor 1-targeted therapy had better to be revised as insulin-like growth factor 1 signaling inhibitors. Targeted therapy also includes IGF1R-activated therapy.

2) In Table 2, the majority of drugs are monoclonal antibodies. Clinical trials of these monoclonal antibodies indicated that ototoxicity should be included in their adverse events. However, the transfer of monoclonal antibodies from circulation to cochlear fluids has not been demonstrated. This is an important issue considering toxicology of these monoclonal antibodies. Therefore, characteristics of pharmacokinetics for monoclonal antibodies should be described in Discussion and stated the need of pharmacokinetic studies for monoclonal antibodies, especially associated with the blood-labyrinth barrier.

Minor critiques

1) In the subchapter ‘Data source’ in Materials and methods, more detailed descriptions about histological findings in the neonatal mouse explant culture model.

2) In the subchapter ‘Systematic review……’ in Materials and methods, pharmacological actions or targets of 17 drugs including search terms should be stated. A table will be desirable.

3) In the line 308 (second paragraph of Discussion), ‘in animal studies and sudden deafness in clinical studies [doi.org/10.1186/1741-7015-8-76; doi.org/10.1186/s12916-014-0219-x]’ is inserted after [48].  
